# A pilot study of neoadjuvant combination of anti-PD-1 camrelizumab and VEGFR2 inhibitor apatinib for locally advanced resectable oral squamous cell carcinoma

Wu-tong Ju [1], Rong-hui Xia[2], Dong-wang Zhu[1], Sheng-jin Dou[1], Guo-pei Zhu [1], Min-jun Dong[3], Li-zhen Wang[2], Qi Sun[3], Tong-chao Zhao[1], Zhi-hang Zhou[1], Si-yuan Liang[1], Ying-ying Huang[1], Yong Tang[1], Si-cheng Wu[4], Jing Xia[5], Shi-qing Chen[5], Yue-zong Bai[5], Jiang Li[2] ✉, Qi Zhu[1] ✉ & Lai-ping Zhong [1,6,7,8] ✉

Novel neoadjuvant therapy regimens are warranted for oral squamous cell carcinoma (OSCC). In this phase I trial (NCT04393506), 20 patients with locally advanced resectable OSCC receive three cycles of camrelizumab (200 mg, q2w) and apatinib (250 mg, once daily) before surgery. The primary endpoints are safety and major pathological response (MPR, defined as ≤10% residual viable tumour cells). Secondary endpoints include 2-year survival rate and local recurrence rate (not reported due to inadequate follow-up). Exploratory endpoints are the relationships between PD-L1 combined positive score (CPS, defined as the number of PD-L1-stained cells divided by the total number of viable tumour cells, multiplied by 100) and other immunological and genomic biomarkers and response. Neoadjuvant treatment is well-tolerated, and the MPR rate is 40% (8/20), meeting the primary endpoint. All five patients with CPS >10 achieve MPR. Post-hoc analysis show 18-month locoregional recurrence and survival rates of 10.5% (95% CI: 0%–24.3%) and 95% (95% CI: 85.4%–100.0%), respectively. Patients achieving MPR show more CD4+ T-cell infiltration than those without MPR (P = 0.02), and decreased CD31 and α-SMA expression levels are observed after neoadjuvant therapy. In conclusion, neoadjuvant camrelizumab and apatinib is safe and yields a promising MPR rate for OSCC.

For patients with locally advanced resectable oral squamous cell carcinoma (OSCC), surgery with adjuvant radiotherapy or chemoradiotherapy has been recommended as the standard treatment[1]. Even after intensive treatments, patients remain at high risk of recurrence or metastasis[2]. In recent years, neoadjuvant therapy before surgery has been shown to reduce the burden of locoregional disease, resulting in improved surgical outcomes; to reduce the risk of distant metastases; and to predict prognosis based on the pathological response in various solid tumours[3]. However, its role in the treatment of OSCC remains ambiguous. Neoadjuvant chemotherapy using cisplatin plus fluorouracil (PF) or docetaxel plus cisplatin plus fluorouracil (TPF) regimens has been explored in patients with OSCC but has not demonstrated survival benefits beyond those provided by standard treatment[4,5]. Thus, exploring effective neoadjuvant therapeutic approaches for locally advanced resectable OSCC remains an urgent need.

Immune checkpoint blockade has been demonstrated to have clinically meaningful antitumor activity in recurrent/metastatic head and neck squamous cell carcinoma (HNSCC, including OSCC)[6,7]. Preclinical data suggest that when the tumour is in place, neoadjuvant immunotherapy stimulates the release of tumour antigens and enhances T-cell priming, thereby resulting in stronger effects than those of adjuvant therapy[8]. In the neoadjuvant setting, immune checkpoint blockade has shown promising results against many other tumour types[9–12]. However, for OSCC or HNSCC, neoadjuvant anti-programmed cell death-1 (PD-1) monotherapy has shown a relatively low major pathological response (MPR) rate (4.3% for pembrolizumab in HNSCC and 8% for nivolumab in OSCC)[13,14].

Targeted drugs against vascular endothelial growth factor receptor (VEGFR) or that inhibit angiogenesis have been shown to relieve immunosuppression through blood vessel normalisation and the oxygen metabolism pathway, thereby having a synergistic effect with anti-PD-1 immunotherapy and concurrently diminishing the risk of immune-related adverse effects[15–18]. The combination of camrelizumab (an anti-PD-1 antibody) and apatinib (a VEGFR inhibitor) has shown favourable antitumor activity and manageable safety in various types of advanced cancers[19–22]. However, this combination has not been studied in patients with locally advanced resectable OSCC.

Here, we show that in patients with locally advanced resectable OSCC, the chemo-free combination of camrelizumab and apatinib as neoadjuvant therapy produces a promising MPR rate with a manageable safety profile.

## Results

### Patient information

From April to December 2020, 21 patients were enroled, and one patient withdrew at the beginning of treatment. The characteristics of the 21 enroled patients are listed in Table 1. Twenty patients received radical surgery, and 18 patients received adjuvant radiotherapy or chemoradiotherapy (Fig. 1).

### Safety

The safety of neoadjuvant camrelizumab and apatinib was evaluated as a primary outcome. The most common neoadjuvant therapy-related adverse events (AEs) were hyperbilirubinemia ($N = 8$, 40%), thrombocytopenia ($N = 7$, 35%) and proteinuria ($N = 6$, 30%). No neoadjuvant therapy-related grade 3 or above AEs were observed (Table 2 and Supplementary Tables 1 and 2). The second cycle of camrelizumab was postponed in one patient for 14 days because of grade 2 thrombocytopenia, and apatinib treatment was suspended in one patient for 21 days because of grade 2 hyperbilirubinemia. Surgery-related AEs, including subcutaneous exudate, posttracheostomy bleeding, postflap-reconstruction pharyngeal fistula, and wound infection, occurred in four patients with one patient per AE. The posttracheostomy bleeding was due to unsecured ligation of the anterior jugular vein, which was detected during the surgical exploration. The other three patients showed no AEs during preoperative laboratory tests, and their surgery-related AEs were all controlled within 2 weeks and were deemed to be unrelated to the neoadjuvant therapy. Two severe AEs occurred: one patient experienced unexplainable elevation in cardiac troponin I levels, which resulted in a surgery delay for 7 days, then recovered within 1 week without any corticosteroid treatment; the other patient experienced unexplained shock after radiotherapy and died.

### Major pathological response

Assessments of the pathological efficacy indicated that MPR was achieved in eight patients (40%, 95% confidence interval [CI]: 19.1–63.9%). The MPR rate in this trial was statistically significantly higher than the null rate of 7% ($p = 0.00003$), meeting the primary endpoint.

**Table 1 | Baseline characteristics**

| Characteristics | N (%) |
|---|---|
| Age, median (range), years | 56.4 (30–71) |
| **Sex** | |
| Male | 12 (57.1) |
| Female | 9 (42.9) |
| **Smoker** | |
| No | 9 (42.9) |
| Yes | 12 (57.1) |
| **ECOG PS** | |
| 0 | 5 (23.8) |
| 1 | 16 (76.2) |
| **Primary tumour site** | |
| Tongue | 6 (28.6) |
| Buccal | 3 (14.3) |
| Gingiva | 5 (23.8) |
| Floor of mouth | 5 (23.8) |
| Palate | 2 (9.5) |
| **Pretreatment clinical T-stage[a]** | |
| T3 | 19 (90.4) |
| T4a | 2 (9.5) |
| **Pretreatment clinical N-stage[a]** | |
| N0 | 16 (66.7) |
| N1 | 3 (14.3) |
| N2 | 2 (9.5) |
| **Pretreatment clinical stage[a]** | |
| III | 17 (80.9) |
| IVA | 4 (19) |
| **Combined positive score** | |
| ≥1 | 16 (76.2) |
| >10 | 5 (23.8) |
| ≥20 | 4 (19) |

[a]American Joint Committee on Cancer, 8th Edition staging.

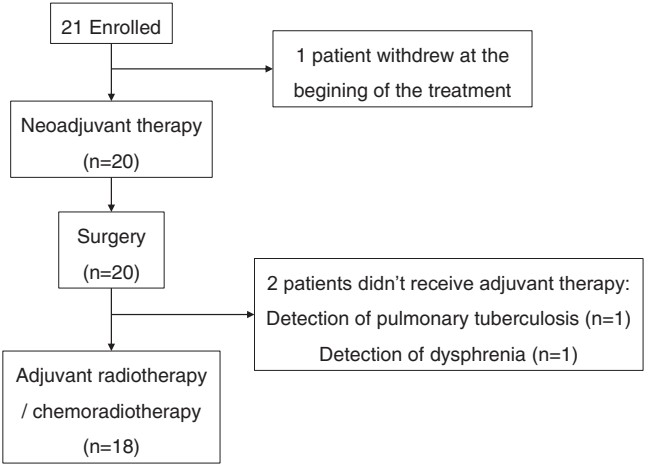

**Fig. 1 | Trial flowchart.** 21 patients were enrolled in this trial; 20 patients received neoadjuvant therapy and surgery; among them, 18 patients received adjuvant therapy.

### Ad hoc radiographic, pathological, and prognostic analyses

Analyses of radiographic responses according to Response Evaluation Criteria in Solid Tumors (RECIST, version 1.1) were performed based on imaging examinations before and after neoadjuvant

**Table 2 | Neoadjuvant therapy-related adverse events (Common Terminology Criteria for Adverse Events Version 5.0) and surgical-related adverse events (Clavien–Dindo) in the 20 patients**

| Adverse event | N (%) | | |
|---|---|---|---|
| | Grade 1 | Grade 2 | Grade ≥3 |
| Skin (rash, dryness, dermatitis) | 0 | 1 (5%) | 0 |
| Pain (lymph node and oral) | 3 (15%) | 1 (5%) | 0 |
| Colitis/Diarrhoea | 1 (5%) | 1 (5%) | 0 |
| Fatigue | 3 (15%) | 0 | 0 |
| Proteinuria | 3 (15%) | 3 (15%) | 0 |
| Hypertension | 1 (5%) | 3 (15%) | 0 |
| Hyperbilirubinemia | 7 (35%) | 1 (5%) | 0 |
| Thrombocytopenia | 6 (30%) | 1 (5%) | 0 |
| Leukopenia | 2 (10%) | 1 (5%) | 0 |
| Increased AST level | 3 (15%) | 0 | 0 |
| Reactive capillary haemangiomas | 3 (15%) | 0 | 0 |
| Surgical toxic effects–Clavien–Dindo scoring | 4 (20%) | 0 | 0 |

*AST* aspartate aminotransferase.

therapy. The radiographic response indicated three patients with partial response (PR), ten patients with stable disease (SD), and six patients with progressive disease (PD) (Supplementary Table 4). One superficial gingival lesion was undetectable on radiographic examinations and thus was not evaluated. Interestingly, among the eight patients who achieved MPR, only three showed radiographic PR (Fig. 2A). One patient with a radiographic PD lesion was further pathologically confirmed to have achieved MPR. All patients with PD lesions received surgery, and no recurrence was observed in primary sites.

For the regional metastatic lymph nodes, a pathological response was observed in 60% (6/10) of patients, with the characteristics of necrosis, multinucleated giant cells, and calcification. In the only patient who achieved pathological complete response (pCR) in the primary tumour, pCR in one lymph node was also observed (Supplementary Table 5).

As of March 2022, the median follow-up time was 18 months (range 15–22 months), and an originally unplanned post hoc analysis of 18-month locoregional recurrence and survival rates was conducted. Two patients who did not receive adjuvant radiotherapy had contralateral neck lymph node metastasis and local recurrence. The estimated 18-month locoregional recurrence rate was 10.5% (95% CI: 0–24.3%). One patient died, and the estimated 18-month overall survival rate was 95% (95% CI: 85.4–100.0%).

## Exploratory analyses of pathological response characteristics

We systematically reviewed the pathological features of resected specimens and proposed immune-related pathological response criteria (irPRC) for neoadjuvant therapy in OSCC. We observed the following characteristics of the immune-related pathological regression bed in OSCC: multinucleated giant cell infiltration, dystrophic calcification, tumour-infiltrating lymphocytes (TIL), foamy macrophages, neovascularization, proliferative fibrosis, tertiary lymphoid structure, and dense plasma cells. In two patients who achieved MPR, tertiary lymphoid structure was observed in the tumours after neoadjuvant therapy (Supplementary Fig. 1).

The association between PD-L1 combined positive score (CPS) and pathological response was examined. All five patients with a CPS value ⟩10 achieved MPR (Fig. 2A and Supplementary Table 3). The number of patients with high CPS showed differences between the MPR and non-MPR groups ($p = 0.004$ for cut-off ⟩10; $p = 0.014$ for cut-off ≥20). One patient with CPS = 90 who achieved radiographic PR was pathologically confirmed to have achieved pCR (Fig. 2B).

Baseline tumour tissues from 15 patients were eligible for next-generation sequencing (NGS). The most frequently mutated gene was *TP53* (14 of 15, 93%), followed by *TERT* (9 of 15, 60%) and *CDKN2A* (6 of 15, 40%) (Fig. 3A). No significant differences in gene mutations, classic pathway enrichment or tumour mutation burden were observed between the MPR and non-MPR groups (Fig. 3B). Multiplex immunofluorescence for TIL staining showed significant increases in the number of CD68+ CD163+ cells ($p = 0.04$) and the CD8+ /FoxP3+ ratio ($p = 0.002$) and decreases in the number of CD3+ ($p = 0.03$) and FoxP3+ ($p = 0.03$) cells from before to after neoadjuvant therapy (Supplementary Fig. 2). The changes in all markers over the course of neoadjuvant therapy were compared between the MPR and non-MPR groups, but no significant differences were observed (Supplementary Fig. 3). The characteristics of TIL infiltration in surgically resected tumours were further compared between the two groups, and no significant difference was found in the levels of

**A**

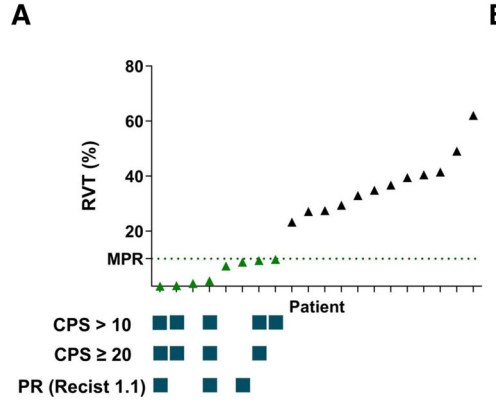

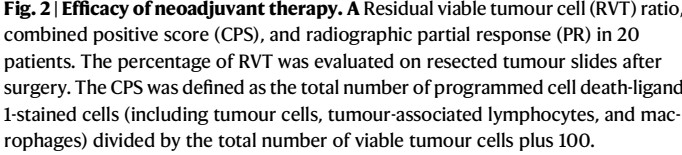

**B**

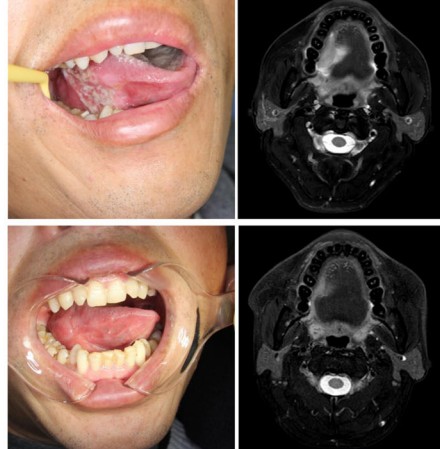

**Fig. 2 | Efficacy of neoadjuvant therapy. A** Residual viable tumour cell (RVT) ratio, combined positive score (CPS), and radiographic partial response (PR) in 20 patients. The percentage of RVT was evaluated on resected tumour slides after surgery. The CPS was defined as the total number of programmed cell death-ligand 1-stained cells (including tumour cells, tumour-associated lymphocytes, and macrophages) divided by the total number of viable tumour cells plus 100.

Radiographic response according to RECIST 1.1 criteria was performed on the basis of imaging examinations before and after neoadjuvant therapy (green triangles for the MPR group [$n = 8$], black triangles for the non-MPR group [$n = 12$]). **B** In the patient who achieved pathological complete response, images of the oral tongue (left) and magnetic resonance imaging (right) before (upper) and after (lower) neoadjuvant therapy are shown. Source data are provided as a Source Data file.

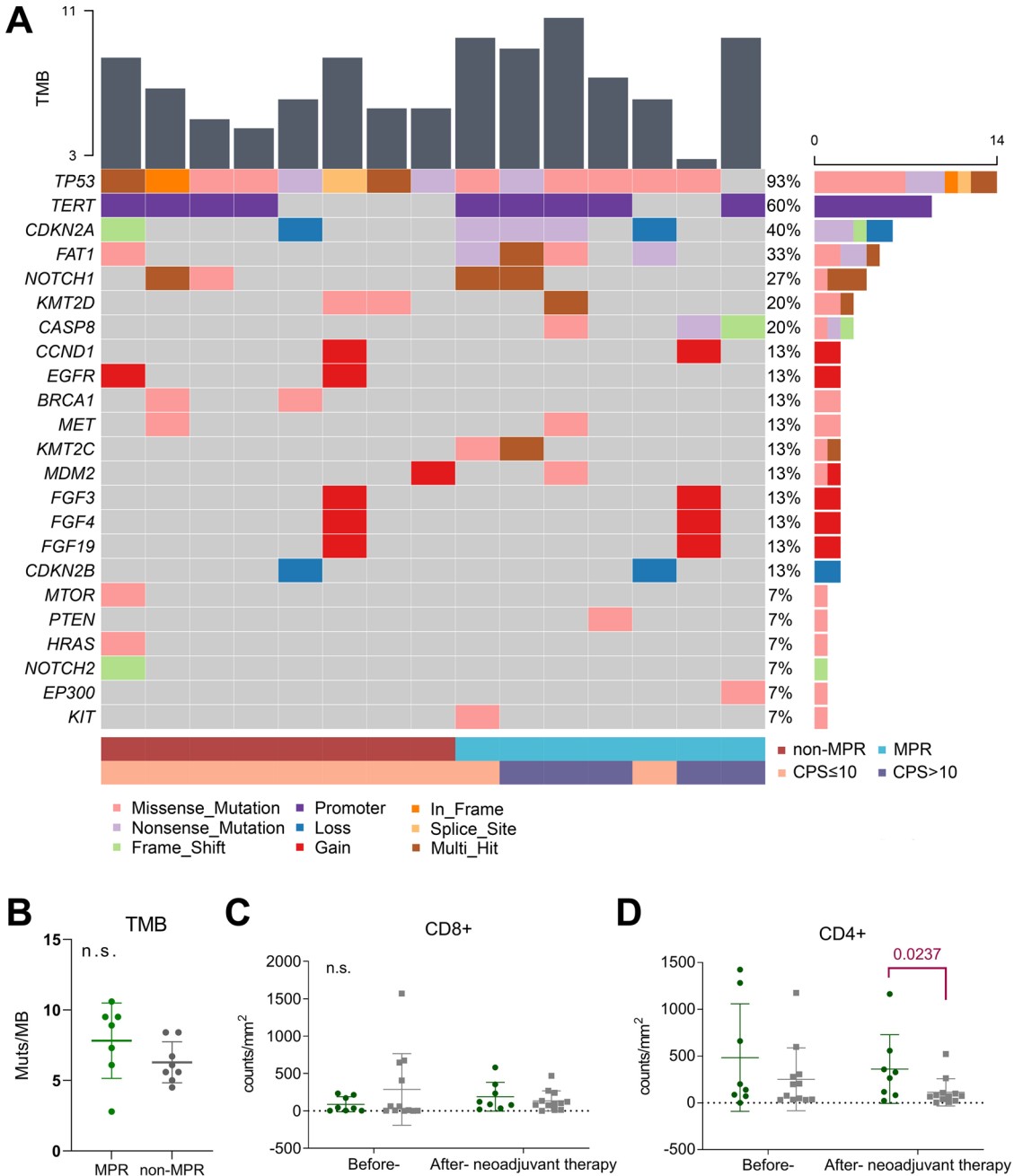

**Fig. 3 | Genetic and tumour-infiltrating lymphocyte analyses. A** Mutations as assessed by next-generation sequencing of baseline primary tumour samples. A column represents a patient. The percentages listed on the right represent the proportion of samples harbouring a mutation in the gene listed on the left. Bottom bars show pathological response (MPR [$n = 8$] or non-MPR [$n = 7$]) and combined positive score ($>$10 [$n = 5$] or $\leq$10 [$n = 10$]) distribution. **B** Comparison of TMB in baseline tumour samples between the MPR and non-MPR groups (green dots for the MPR group [$n = 7$], grey dots for the non-MPR group [$n = 8$]). Quantitative graphs of the infiltration density of CD8+ (**C**) and CD4+ (**D**) T cells in tumour samples before and after neoadjuvant therapy (green dots for the MPR group [$n = 8$], grey dots for the non-MPR group [$n = 12$]). The significance of the differences between before and after neoadjuvant therapy was tested using a two-sided Wilcoxon signed-rank test; for differences between the MPR and non-MPR groups, the significance was tested using a two-sided Mann–Whitney test. Bars represent the mean with SD. Source data are provided as a Source Data file.

CD8+ T-cell infiltration (Fig. 3C), but the MPR group showed higher levels of CD4+ T-cell infiltration ($p = 0.02$, Fig. 3D and Supplementary Fig. 4).

One patient from the non-MPR group was found to exhibit tumour progression on radiographic evaluation, with a change in growth kinetics exceeding 50% and new neck lymph node metastasis, and was confirmed to exhibit disease hyperprogression (HPD, Fig. 4A, B). Interestingly, for this patient, high levels of CD8+ cell and CD163+ cell infiltration were found at baseline, and the levels of

CD8+ cells were diminished while the levels of CD163+ cells were significantly elevated in the surgical sample (Fig. 4C, D and Supplementary Fig. 2).

During the evaluation of angiogenesis, decreased CD31 (a marker of vascular endothelial cells) and α-SMA (a marker of pericytes) expression levels were observed after neoadjuvant therapy, thus confirming the antiangiogenic effect in tumours (Fig. 5A, B). No significant difference in CD31 or α-SMA expression was found between the MPR and non-MPR groups (Fig. 5C).

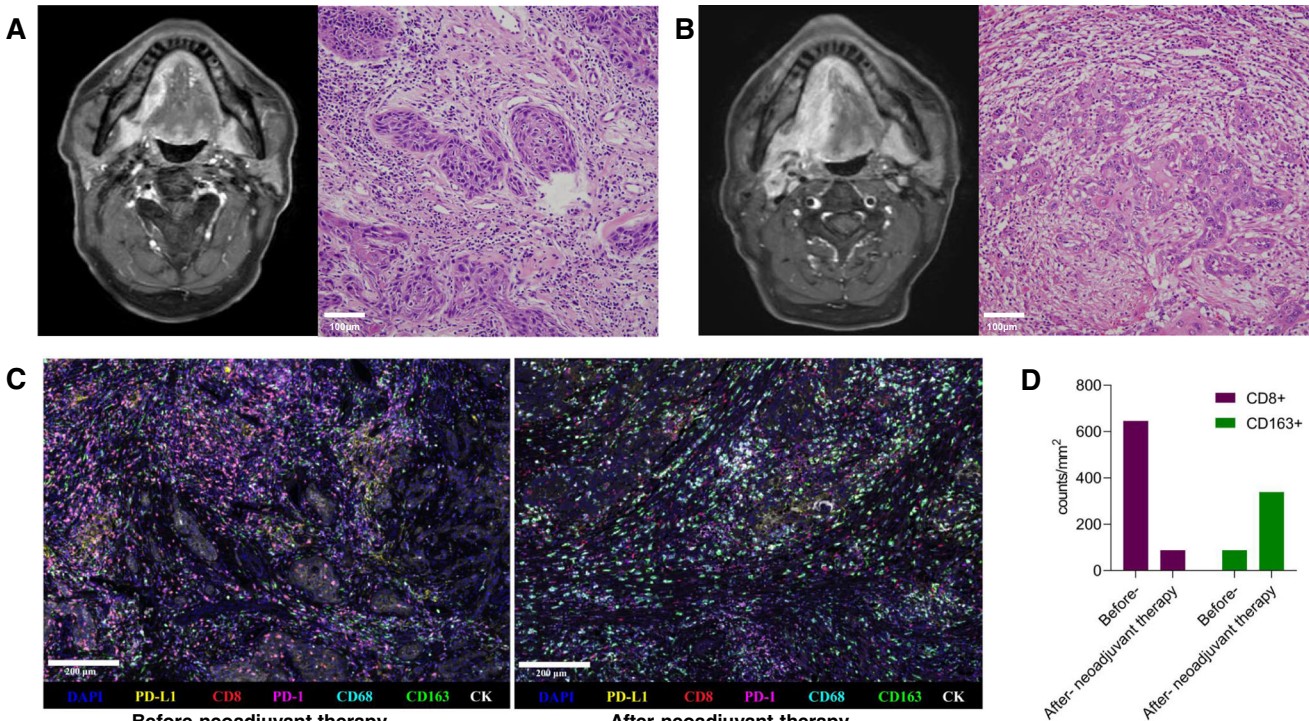

**Fig. 4 | Features of the hyperprogressive disease.** In patient No. 14, who showed hyperprogressive disease: radiographic and H&E staining images before (**A**) and after (**B**) neoadjuvant therapy. **C** Multiplex immunofluorescence images of the tumour site before and after neoadjuvant therapy. Primary antibodies targeting CD163, CD68, PD-1, CD8, PD-L1, and Pan-CK were used. Nuclei acids were stained with DAPI. **D** Comparison of fluorescence intensity for CD8+ and CD163+ cells. After staining, slides were scanned, and multilayer images were used for quantitative image analysis. The quantities of various cell populations were expressed as the number of stained cells per square millimetre in all nucleated cells. Due to limited tumour tissue obtained by biopsy, immunofluorescence experiments were performed without repetition. Source data are provided as a Source Data file.

## Discussion

Our results provide the first evidence that neoadjuvant therapy using a chemo-free combination of camrelizumab and apatinib is well tolerated in patients with OSCC, with an MPR of 40%.

The safety profile of camrelizumab and apatinib in the neoadjuvant setting was mostly consistent with that previously reported in advanced cancers[19,20,23]. We observed no grade 3–4 neoadjuvant therapy-related AEs, which might be due to the short course of camrelizumab and apatinib administration. Furthermore, the safety that was observed in our trial seems to be superior to that observed in other neoadjuvant chemotherapy regimens used in OSCC trials, such as neoadjuvant chemotherapy with the TPF regimen (9–38% grade 3–4 therapy-related AEs)[4,24,25] or targeted therapies (25–61.6% grade 3–4 therapy-related AEs)[26,27]. Recent neoadjuvant immunotherapy studies in head and neck cancer showing favourable side effects further supported the superior safety of neoadjuvant immunotherapy[13,28,29]. Since previous studies reported that anti-VEGF(R) therapies could increase the risk of bleeding[30,31], we set a time interval of 5 days between apatinib administration and surgery to reduce the risk of complications during subsequent surgery. Although the surgery-related AEs observed in our study were considered unrelated to neoadjuvant therapy, trials with larger sample sizes are necessary to definitively indicate the effects of neoadjuvant anti-PD-1 plus anti-VEGFR therapy on surgery.

In addition to safety, pathological efficacy, including MPR, is a crucial criterion for proposing neoadjuvant therapy in OSCC or HNSCC. The combination of camrelizumab and apatinib showed a promising MPR (40%), as compared with chemotherapy with PF (33%) or TPF (27.7%) in OSCC[4,25], pembrolizumab monotherapy (4.3–20.5%) and nivolumab monotherapy (5.9%) in HPV-unrelated HNSCC[14,29,32], and nivolumab monotherapy (8%) or nivolumab combined with ipilimumab (20%) in OSCC, although cross-study comparisons should be made with caution[13]. In another neoadjuvant therapy trial for HNSCC, the cisplatin/docetaxel/durvalumab/tremelimumab combination in a neoadjuvant setting showed a superior pathological response in terms of pCR (48%) but also had a higher rate of grade 3–4 AEs (68%)[33]. One trial using immune radiotherapy in a neoadjuvant setting achieved a high MPR rate (86%) in HNSCC, thus supporting the further evaluation of the addition of stereotactic body radiation therapy to neoadjuvant immunotherapy[34]. The results from this study suggested that a high CPS might predict MPR from neoadjuvant therapy with camrelizumab and apatinib. Due to the limited sample size, the predictive value of CPS for anti-PD-1 plus anti-VEGFR therapy that was observed in this study must be validated in a larger study. The concept of using CPS to guide the choice of neoadjuvant immunotherapy was consistent with principles suggested for recurrent/metastatic HNSCC[35].

In all the reported neoadjuvant immunotherapy trials for OSCC or HNSCC, the pathological response varied in terms of MPR ratios, and the assessment procedures also varied. Unlike in non-small cell lung carcinoma and melanoma[36,37], in OSCC or HNSCC, irPRC have not been well defined. In previous neoadjuvant immunotherapy trials, the pathological response has been described as featuring a "visible regressed tumour" in addition to "inflammation, giant cell reaction and acellular keratin" and has been quantified as occurring in "a percentage of the overall tumour bed (area of pathological response/area of pathological response plus viable tumour)"[13,29]. Regardless of the assessment method that is used, the definition of the tumour regression bed after neoadjuvant therapy is key, especially in tumours that have significantly shrunk. Based on the criteria for determining the range of the immunotherapy-induced tumour regression bed proposed in lung cancer[37], we systematically

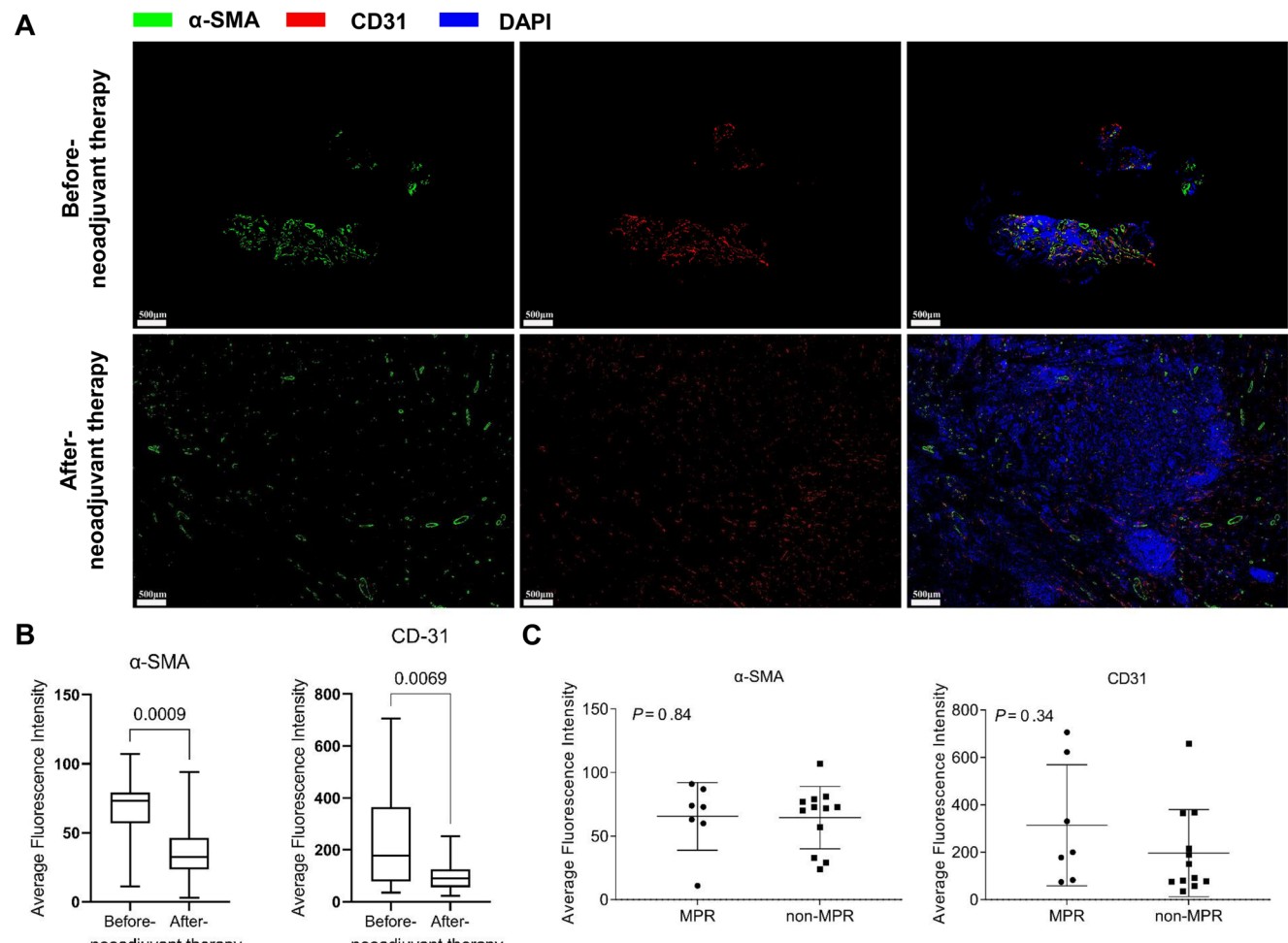

**Fig. 5 | Anti-angiogenesis evaluation. A** Representative immunofluorescence staining images of before and after neoadjuvant therapy tumour sections (Case No. 7, green for CD31, red for α-SMA, blue for DAPI). **B** Fluorescence intensity of CD31 and α-SMA expression before and after neoadjuvant therapy tumour tissues in all 20 patients. Whiskers represent min to max. Bounds of boxes represent 25th and 75th percentiles, centres represent medians, whiskers represent min to max. The significance for differences between before- and after neoadjuvant therapy was tested using a two-sided Wilcoxon signed-rank test. **C** Comparison of baseline CD31 and α-SMA fluorescence intensity between the MPR and non-MPR groups ($n = 8$ in the MPR group, $n = 12$ in the non-MPR group). The significance of the differences between the MPR and non-MPR groups was tested using a two-sided Mann–Whitney test. Bars represent the mean with SD. Source data are provided as a Source Data file.

evaluated the features of tumours from different oral cavity sites in this trial, thus providing a reference for irPRC for subsequent neoadjuvant immunotherapy in OSCC.

The irPRC we proposed for OSCC did not include the pathological response characteristics for neck lymph node metastasis. We found only one lymph node with confirmed tumour regression, whereas a different reaction occurred in another lymph node from the same patient. As described in a previous lung cancer study, the limitations of nodal disease assessment have been attributed to sampling issues[37]. In agreement with findings from a study using nivolumab, a similar response was found between primary sites and lymph nodes[38]. Nodal upstaging occurred in four patients in our study, and all the corresponding primary tumours did not achieve MPR; therefore, this finding suggests that lymph nodes should be monitored more frequently during neoadjuvant treatment.

Because of the short period of 2 weeks between the last neoadjuvant immunotherapy and surgery, radiographic re-evaluation based on modified RECIST 1.1 criteria for immune-based therapeutics (iRECIST) could not be performed in this study[39]. In pathological re-evaluation of the resected lesions, the RECIST 1.1 criteria did not show sufficient sensitivity in response assessment in our trial. Among the eight patients who achieved MPR, only three

showed PR on radiographic scans. One patient with a radiographic PD lesion in our trial was further pathologically confirmed to have achieved MPR, thus indicating the importance of the re-evaluation of progression after immunotherapy. This finding was consistent with the radiographic response analysis of neoadjuvant immunotherapy in lung cancer, in which 30% of patients with SD nearly achieved pCR[40]. In future neoadjuvant therapy trials, other modified methods for radiographic response evaluation should be proposed, such as the criteria used in the window of opportunity, in which a size reduction >10%, rather than 30%, might be defined as indicating a "radiographic responder"[38].

Consistent with previous reports[41,42], *TP53* was found to be the most frequently mutated gene in our study. In agreement with findings from other neoadjuvant immunotherapy trials for OSCC or HNSCC[13,29,42], neither the mutated gene enrichment nor the degree and features of baseline TIL infiltration predicted MPR in our study. The limited sample size and the heterogeneity between biopsy and surgically resected tissues might be the reason, although we matched the biopsy and surgically resected tissues by the clinical tumour sites and features, as well as the microscopic tumour cell proportion. However, higher levels of CD4+ T-cell infiltration were observed in resected tumours that achieved MPR. However, an abnormal increase in the

level of CD163+ cell infiltration was observed in HPD tumours. A higher level of CD4+ cells was associated with better outcomes of neoadjuvant therapy, and M2 macrophages in the tumour microenvironment have been reported to be associated with the occurrence of HPD[43,44]. Further analyses of our tumour samples are urgently needed to explore the underlying mechanisms.

The expression of angiogenesis markers following apatinib treatment, including CD31 and α-SMA, was inhibited by neoadjuvant therapy in this trial, which is similar to previously reported preclinical results[45,46]. However, the expression of angiogenesis markers showed no differences between the MPR and non-MPR groups, and this paradoxical finding might be due to the small sample size of our trial. However, the multiple steps of the cancer immunity cycle and multiple signalling pathways have been found to be affected when combining antiangiogenic agents with anti-PD-1 therapy. Antiangiogenic therapy can inhibit angiogenic signalling and result in normalisation of the tumour vasculature to diminish the immunosuppression exerted by immunosuppressive cells (e.g., Tregs, tumour-associated macrophages, and myeloid-derived suppressor cells), as well as inhibit the expression of PD-1 and regulate apoptotic pathways in cytotoxic CD8+ T cells, thereby improving the efficacy of immunotherapy[47,48]. Huang et al. also reported that crosstalk between tumour vascular normalisation and immune reprogramming pathways exists and plays a vital role in responses[49]. In addition, the LTβR signalling pathway and angiogenic pathway ANGPT2/Tie2 have been reported to be modulated after combinatory treatment with an antiangiogenic agent and anti-PD-1 antibody[50]. In our trial, whether apatinib affects other pathways or other cell types when combined with camrelizumab deserves further investigation.

The limitations of this trial mainly include the small sample size and single-arm design, which lacked a control arm. Nevertheless, the primary endpoint MPR rate was promising in spite of small sample size. It is important to emphasise that although MPR or even pCR have been considered candidate early surrogate endpoints for survival in the neoadjuvant setting[36,51,52], whether they might result in long-term survival improvement remains to be confirmed in this and other neoadjuvant immunotherapy trials. In addition, since sufficient baseline tumour tissues were not available for all patients, definitive conclusions regarding biomarkers remain ambiguous. Further in-depth analysis of biomarkers in patient tumours and blood (such as the detection of PET-CT parameters, analyses of tumour mutational signatures, RNA analyses, and analyses of neoantigens, circulating tumour DNA, and T-cell activation and exhaustion) should be performed in our ongoing randomised trial (NCT05069857) and other ongoing neoadjuvant trials[51].

In conclusion, this pilot trial showed that neoadjuvant therapy using a chemo-free combination of camrelizumab and apatinib was well tolerated in patients with OSCC. The MPR rate was promising, and CPS might be a signal predictor. These results suggest that further neoadjuvant therapy trials for OSCC using anti-PD-1 plus anti-VEGFR should be conducted.

## Methods

### Study population and trial design

This single-centre, open-label phase I trial was performed at the Ninth People's Hospital, Shanghai Jiao Tong University School of Medicine in Shanghai, China. Eligible patients were aged 18–75 years, had histopathologically confirmed locally advanced OSCC with a clinical stage of III or IVA (American Joint Committee on Cancer, 8th Edition), and had an Eastern Cooperative Oncology Group performance status of 0 to 2. Patients were enroled between April 2020, and December 2020. The full eligibility criteria are provided in the trial protocol (Supplementary Note 1).

The trial followed the ethical guidelines of the Declaration of Helsinki and was approved by the Institutional Ethics Committee,

Ninth People's Hospital, Shanghai Jiao Tong University School of Medicine. Each patient provided signed informed consent before participating in this trial. The authors affirm that human research participants provided informed consent for publication of the images in Fig. 2B. An authorisation of the release of the data was obtained from the Clinical Research Broad of Ninth People's Hospital. This trial was registered on ClinicalTrials.gov (NCT04393506), which was submitted on May 3, 2020, after the enrolment of one patient. Since the coronavirus disease-19 (COVID-19) outbreak, there has been a large backlog in clinical work, resulting in a delay in the registration of our trial. In this study, camrelizumab and apatinib were free for patients.

### Procedures

The patients received three cycles of intravenous camrelizumab (200 mg) on d1, d15, and d29 and oral apatinib (250 mg) daily, starting on d1 and ending on the 5th day before surgery. Standard radical surgery was planned on d42–45. Adjuvant radiotherapy or chemoradiotherapy was planned within 6 weeks after surgery, according to the pathological stage.

The standard operation procedure of determining the regression bed induced by neoadjuvant immunotherapy in oral cancer was proposed. Briefly, before neoadjuvant therapy, the baseline tumour bed was recorded by photographing and radiographic examination. Marks using tattoo or methylene blue were placed 0.5 cm from the palpable margins of the tumour. After neoadjuvant therapy, features of the tumour were also recorded before surgery. The resection range was determined by the baseline tumour bed even if the tumour shrank after neoadjuvant therapy. In this case, we connected the original marking points as a reference for the determination of surgical safety margins (0.5–1.0 cm away from the marking points) and pathological tumour bed.

According to a previous study[52], haematoxylin and eosin-stained (H&E) slides of sections of the residual tumour were assessed by pathologists blinded to the patient information. Slices of at least 1 section per 3 mm of the greatest tumour diameter were obtained during sampling. The standard operating procedure of slide preparation is shown in Supplementary Figs. 5 and 6.

The percentage of residual viable tumour (RVT) cells was evaluated by H&E staining of all slides. According to a previous study[37], the tumour bed was defined as the areas of "residual viable tumour + necrosis + regression bed" (Supplementary Table 6). The definition and characteristic images of the immune-related pathological tumour regression bed in OSCC are shown in Supplementary Table 7 and Supplementary Figs. 1, 7–9. RVT% was determined by summing all tumour areas and then dividing by the sum of all tumour bed areas in all slides. Radiographic response evaluation was performed according to the RECIST 1.1 criteria. Data were collected using Microsoft Office Excel 2019.

### Study endpoints

The primary endpoints were safety and MPR rate (MPR, defined as the presence of 10% or fewer RVT cells). The 2-year survival rate (defined as the proportion of patients alive at the 2-year follow-up) and local recurrence rate (defined as the proportion of patients with local recurrence) were the secondary endpoints, which were not reported due to inadequate follow-up. AEs were assessed according to the Common Terminology Criteria for Adverse Events (version 5.0). Neoadjuvant therapy-related AEs were managed mainly according to the American Society of Clinical Oncology Clinical Practice Guidelines[53]. Surgery-related AEs were assessed using the guidelines of the Clavien–Dindo Classification of Surgical Complications[54].

### Immunohistochemistry and multiplex immunofluorescence

Programmed cell death-ligand 1 (PD-L1) expression was evaluated with a PD-L1 immunohistochemistry (IHC) 22C3 pharmDx assay (Dako

North America, Carpinteria CA). CPS was defined as the total number of PD-L1-stained cells (including tumour cells, tumour-associated lymphocytes, and macrophages) divided by the total number of viable tumour cells, multiplied by 100.

The Akoya OPAL Polaris 7-Colour Automation IHC kit (NEL871001KT) was used to evaluate TIL. Formalin-fixed, paraffin-embedded (FFPE) tumour slides were deparaffinized in a BOND RX system (Leica Biosystems) and then sequentially incubated with primary antibodies targeting CD163 (Abcam, ab182422, 1:500), CD68 (Abcam, ab213363, 1:1000), PD-1 (CST, D4W2J, 86163S, 1:200), CD3 (Dako, A0452), CD4 (Abcam, ab133616, 1:100), CD8 (Abcam, ab178089, 1:100), CD56 (Abcam, ab75813, 1:100), CD20 (Dako, L26, IR604), FOXP3 (Abcam, ab20034, 1:100) and Pan-CK (Abcam, ab7753, 1:100) (Akoya Biosciences). Then, the cells were incubated with secondary antibodies and corresponding reactive Opal fluorophores. Nuclei acids were stained with DAPI. Slides incubated with primary and secondary antibodies without fluorophores were used as negative controls.

After staining, slides were scanned using a Vectra Polaris Quantitative Pathology Imaging System (Akoya Biosciences) at 20 nm wavelength intervals from 440 nm to 780 nm with a fixed exposure time and an absolute magnification of ×200. All scans for each slide were then superimposed to obtain a single image. Multilayer images were imported into inForm v.2.4.8 (Akoya Biosciences) for quantitative image analysis. The tumour parenchyma and stroma were differentiated by Pan-CK staining. The quantities of various cell populations were expressed as the number of stained cells per square millimetre in all nucleated cells.

CD31 (tumour endothelial cells) and α-SMA (pericytes) staining were performed to evaluate vascular normalisation. The antibodies used were anti-CD31 #ab178981 (dilution 1:1000, Abcam, USA), Alexa Fluor 594 donkey anti-rabbit lgG(H + L) #A21207 (dilution 1:400, Life Technologies, USA), anti-alpha smooth muscle actin [1A4] #ab7817 (dilution 1:10,000, Abcam, USA) and Alexa Fluor 488 donkey anti-mouse lgG(H + L) # A21202 (dilution 1:400, Life Technologies, USA) in addition to DAPI (dilution 1:500, #C0060, Solarbio, CHN). The obtained slices were observed by confocal fluorescence microscopy (Leica SP5, Germany). The fluorescence intensity was analysed by ImageJ software.

### Targeted NGS and genetic analysis

Formalin-fixed paraffin-embedded tissue sections were evaluated for tumour cell content using H&E staining. Only samples with a tumour content of ≥20% were eligible for subsequent analyses. Genomic DNA was isolated from tissue samples using the ReliaPrep™ FFPE gDNA Miniprep System (Promega) and quantified using the Qubit™ dsDNA HS Assay Kit (Thermo Fisher Scientific) following the manufacturer's instructions. DNA extracts (30–200 ng) were sheared to 250 bp fragments using an S220 focused ultrasonicator (Covaris). For targeted capture, indexed libraries were subjected to probe-based hybridisation with a customised NGS panel targeting 733 cancer-related genes.

The captured libraries were loaded onto a NovaSeq 6000 platform (Illumina) for 100 bp paired-end sequencing with a mean sequencing depth of 1000. Tumour mutational burden was defined as the number of nonsynonymous somatic single nucleotide variants (SNVs) and indels in examined coding regions, with driver mutations excluded. All SNVs and indels in the coding region of targeted genes, including missense, silent, stop gain, stop loss, in-frame and frameshift mutations, were considered. The "maftools" package was used to examine the genomic landscape. Copy number variation analysis was performed using an in-house developed pipeline. A fold-change threshold of 1.6 and 0.6 in DNA copy number was set as the cut-off for amplification and deletion, respectively. The key pathway-related genes were visualised, including those involved in *HGF* signalling,

*EGFR*/*RAS*/*BRAF* signalling, *CDK* signalling, *AKT*/*mTOR*/*PI3K* signalling, *FGFR* signalling, *p53* signalling, epigenetic/chromatin remodelling, DNA damage and repair/telomere stability, and *NOTCH* signalling. Other clinical trial drug target and TERT promoter hot spot mutations were also shown in the genomic landscape.

### Statistical analysis

A sample size of 20 evaluable patients was required to achieve 90% power to detect an increase in the MPR rate from 7% (anti-PD-1 monotherapy, based on data from pembrolizumab and nivolumab monotherapy[13,14]) to 30% using a one-sided exact test with a significance level (alpha) of 0.050.

In an unplanned post hoc analysis, the 18-month overall survival rate and local recurrence rate was evaluated with the Kaplan–Meier method. The CIs were calculated with the Brookmeyer–Crowley method for survival and recurrence rates and calculated with the Clopper–Pearson method for the MPR rate. The $p$ value comparing the observed MPR rate to the null rate was calculated with Fisher's exact test. Based on the different CPS cut-offs, the $p$ value of the difference in the number of patients between the MPR and non-MPR groups was calculated with Fisher's exact test. Differences between the MPR and non-MPR groups were analysed using the Mann–Whitney test, and differences in the measured indicators before and after neoadjuvant therapy within a group were analysed using the Wilcoxon signed-rank test. The significance level for two-sided $p$ values was set at 0.05 in statistical analyses. Statistical analysis was performed in IBM SPSS Statistics (version 23), GraphPad Prism software (version 9.1.2) and SAS (version 9.4).

### Reporting summary

Further information on research design is available in the Nature Research Reporting Summary linked to this article.

### Data availability

The trial protocol is available as Supplementary Note 1 in the Supplementary Information file. Deidentified clinical data (including safety, radiographic and pathological analyses of neoadjuvant therapy, demographic, tumour-related clinical information) of each patient underlying the results reported in this manuscript and multiplex immunofluorescence data are available in the manuscript or additional files. Other deidentified data, such as radiograpic and pathological images, blood test results of each patient, can be obtained for research purposes from the corresponding author at zhonglp@hotmail.com. DNA data have been deposited in the Genome Sequence Archive for Human (https://ngdc.cncb.ac.cn/gsa-human/) under accession codes for HRA002175. Due to the policy of our hospital, the approval from the Clinical Research Unit needs to be obtained before the release of patients' DNA data. Access to DNA data can be obtained for research purposes from the corresponding author at zhonglp@hotmail.com, who will contact the Clinical Research Unit, Ninth People's Hospital, College of Stomatology, Shanghai Jiao Tong University School of Medicine. Once the access has been granted, the data will be permanently available for the requester. The remaining data are available within the Article, Supplementary Information or Source Data file. Source data are provided with this paper.

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

## Acknowledgements

We thank Ding Zhang and Bin Li for their hard work in performing DNA sequencing and analysis. We thank Lei Li for her hard work in the collection and preservation of pathological samples. We thank Xi-yu Zheng for her hard work in performing IHC. We thank Wei Zhao and Qian-qian Li for their hard work as CRC. We thank Xiao-yue Wu and Yan-hua Xu from Jiangsu Hengrui Pharmaceuticals Co., Ltd. for medical writing assistance during major revision. We thank all patients and their relatives who participated in this clinical trial. This work was supported by the National Natural Science Foundation of China (grant numbers 82172734 (L.Z.), 82103043 (W.J.), 81972525 (L.Z.)), the projects from Science and Technology Commission of Shanghai Municipality (grant numbers 21Y21900300 (L.Z.), 19XD1422300 (L.Z.)), and Ninth People's Hospital, Shanghai Jiao Tong University School of Medicine (grant numbers 201916 (L.Z.), YBKB202111 (L.Z.)). The funding sources had no role in the design of the study and collection, analysis, and interpretation of data or in writing the manuscript.

## Author contributions

L.Z. was the chief investigator of the trial. L.Z., Q.Z., and J.L. contributed to the conception and design of the trial. W.J., R.X., D.Z., S.D., G.Z., M.D., L.W., Q.S., T.Z., Z.Z., S.L., Y.H., Y.T., J.X., S.C., Y.B., J.L., Q.Z., and L.Z. contributed to patient enrolment and care and the acquisition of data. W.J., R.X., and D.Z. contributed to data analysis, data interpretation, and drafting of the manuscript. W.J., R.X., and S.W. contributed to the statistical analysis. W.J., R.X., and D.Z. contributed equally to this paper. All authors read and approved the final manuscript.

## Competing interests

J.X., S.C. and Y.B. are employees of the company 3D Medicines Inc. The remaining authors declare that they have no competing interests.

## Additional information

[1]Department of Oral and Maxillofacial-Head and Neck Oncology, Ninth People's Hospital, College of Stomatology, Shanghai Jiao Tong University School of Medicine, Shanghai 200011, PR China. [2]Department of Oral Pathology, Ninth People's Hospital, College of Stomatology, Shanghai Jiao Tong University School of Medicine, Shanghai 200011, PR China. [3]Department of Radiology, Ninth People's Hospital, College of Stomatology, Shanghai Jiao Tong University School of Medicine, Shanghai 200011, PR China. [4]Biostatistics Office of Clinical Research Unit, Ninth People's Hospital, College of Stomatology, Shanghai Jiao Tong University School of Medicine, Shanghai 200011, PR China. [5]The Medical Department, 3D Medicines Inc., Shanghai 200011, PR China. [6]National Center for Stomatology, Shanghai 200011, PR China. [7]National Clinical Research Center for Oral Diseases, Shanghai 200011, PR China. [8]Shanghai Key Laboratory of Stomatology, Shanghai 200011, PR China. ✉e-mail: lijiang182000@126.com; zhuqi70@hotmail.com; zhonglp@hotmail.com

