## [Peer Review File · Nature Communications]

Reviewers' Comments:

Reviewer #1:

Remarks to the Author:

In this clinical trial testing the value of PD1 and VEGF blockade in patients with advanced oral SCC in the Neo-adjuvant (pre-surgical) setting, the authors show remarkable response in 40% of the 20 patients who undergo the planned treatment, and this appears to be enriched in patients with CPS >10, potentially establishing an important role for this combination albeit in a small study population. I have a number of major issues that need to be addressed in this study.

1. While this is indeed a commendable study (I understand the difficulties of running Neo-adjuvant studies on surgical patients), it is a small study in terms of patient numbers, and hence I am finding it challenging to either place it as a purely clinical Phase 2 study (where the numbers here are too low to make a clinical impact) versus a biomarker/tumor analyses study- where the depth of the analysis performed is very limited!

2. If indeed the authors are positioning this manuscript as a biomarker analysis paper of a well-curated, tight but small dataset, then I would expect to see greater depth in the analyses, and these should include the following:

- DNA analyses to look at tumor mutation burden, genomic landscape and mutational signatures
- RNA analyses to interrogate the transcriptome pre and post treatment, and provide some additional support for the multiplex immune-fluorescence shown here
- if RNA analyses is not possible, greater depth in biomarker analysis by Vectra stain as demonstrated here, but including a range of markers, encompassing T-cell dysfunctional markers (LAG3, TIM3, VTLA4 etc), lineage subsets (CD103, CD39, CD107 etc), macrophage markers, downstream effectors (GZMB, PRF etc) amongst others.

3. Failing to do 2, results in a paper where the main material is highlights of a few interesting cases amongst the 8 responders.

4. I am surprised to see that PD1 staining did not change after treatment with PD1-blockade? would there not be occupancy of the receptor preventing binding of the IHC antibody?

5. while the angiogenic markers are changing, it does not appear to be affecting their correlation with response, is the antiangiogenic drug affecting other pathways or other cell types such as macrophages etc

Minor issues that need to be addressed:

1. many typographical errors that need to be corrected, including those in the main figures (eg Fig 1- surgery and radiotherapy have been misspelled)

2. missing legends for many of the graphs (eg Figure 2, 4, 5).

3. Statistical tests used should also be indicated in the figure legends.

Reviewer #2:

Remarks to the Author:

I would like to commend the authors on a well put together manuscript. Below are a few minor comments/suggestions:

1. For the study endpoints, are all AEs reported or only treatment related AEs?

2. In the study design, is there a reference for the MPR rate of 7%? This should potentially be mentioned in the background as well.

3. When comparing the fluorescence using the t-test, was normality assessed? Many of the markers appear as though they may be skewed.

4. In figure 3, the color code does not match the legend (images have green/gray not blue/red).

4a. In figure 3, were comparisons made between pre-post within a group? It might also be worthwhile to compare the changes in these markers between groups.

4b. Please provide the actual p-values for significant results.

5. if not all patients have adequate follow-up (ie. 12 months), then is the clopper-pearson method appropriate for estimating the 12-month survival and local recurrence rates? These should be estimated off of Kaplan-Meier curves.

6. In the efficacy results, a confidence interval for the MPR rate should be provided, along with the exact test p-value comparing the observed rate to the null rate of 7%.

Reviewer #1 (Remarks to the Author):

In this clinical trial testing the value of PD1 and VEGF blockade in patients with advanced oral SCC in the Neo-adjuvant (pre-surgical) setting, the authors show remarkable response in 40% of the 20 patients who undergo the planned treatment, and this appears to be enriched in patients with CPS >10, potentially establishing an important role for this combination albeit in a small study population. I have a number of major issues that need to be addressed in this study.

1. While this is indeed a commendable study (I understand the difficulties of running Neo-adjuvant studies on surgical patients), it is a small study in terms of patient numbers, and hence I am finding it challenging to either place it as a purely clinical Phase 2 study (where the numbers here are too low to make a clinical impact) versus a biomarker/tumor analyses study- where the depth of the analysis performed is very limited!

Reply: Thank you for your valuable comment. Considering that the sample size of our study was small, we initially positioned this study as a pilot study. This type of studies generally requires a quite small sample size and aims to provide feasibility data for subsequent large studies by exploring and assessing the feasibility of a new/experimental treatment or new therapeutic combination/regime.^{1,2} In addition, DNA analyses was performed as per your suggestion and the results have been presented in the revised manuscript (page 8, paragraph 3, and Fig 3A, B):
Currently, we are conducting a further randomized controlled phase II trial to verify the value of neoadjuvant chemo-free combination of camrelizumab and apatinib and focus on a more in-depth biomarker analysis for the selection of the beneficiary population (NCT05069857).

Reference:

1. Thabane L, et al. A tutorial on pilot studies: the what, why and how. BMC Med Res Methodol 10, 1 (2010).
2. Hee SW, et al. Decision-theoretic designs for small trials and pilot studies: A

review. *Stat Methods Med Res* 25, 1022-1038 (2016).

2. If indeed the authors are positioning this manuscript as a biomarker analysis paper of a well-curated, tight but small dataset, the I would expect to see greater depth in the analyses, and these should include the following:

- DNA analyses to look at tumor mutation burden, genomic landscape and mutational signatures
- RNA analyses to interrogate the transcriptome pre and post treatment, and provide some additional support for the multiplex immune-flourescence shown here
- if RNA analyses is not possible, greater depth in biomarker analyse by Vectra stain as demonstrated here, but including a range of markers, encompassing T-cell dysfunctional markers (LAG3, TIM3, VTLA4 etc), lineage subsets (CD103, CD39, CD107 etc), macrophage markers, downstream effectors (GZMB, PRF etc) amongst others.

Reply: Thank you for your comment. We performed next-generation sequencing (NGS) with a customized panel targeting 733 cancer-related genes in 15 cases with eligible biopsy tissue (the detailed methods are shown in Additional file 3), as per your suggestion. The results showed that the most frequently mutated gene was TP53 (14 of 15, 93%), followed by TERT (9 of 15, 60%) and CDKN2A (6 of 15, 40%). No significant difference was found for gene mutation or classic pathway enrichment or tumor mutation burden (TMB) between MPR and non-MPR groups. The above results have been presented in the revised manuscript (page 8, paragraph 3, and Fig 3A, B).

For mutational signatures analysis, it generally requires more than 50 mutations in each case, which is usually obtained by whole exome sequencing (WES). Instead of WES, we used a customized panel targeting 733 genes, and there were relatively few mutations in each sample. Bias could not be overcome in the process of bioinformatics analysis for mutational signatures. Moreover, the baseline tumor tissue obtained by biopsy is inadequate for DNA analyses of the other 5 cases and other analyses such as RNA, further immune-flourescence analyses, to allow comparison

with the post-treatment data. We believe that your concern is of importance for our further study and we plan to perform these analyses that you suggest in the subsequent trial (NCT05069857). Many thanks for your kind help!

We added this point of view in the Discussion part of the revised manuscript (page 15, paragraph 1) as: To investigate the predictive biomarkers for refined neoadjuvant therapy, further in-depth biomarkers in the patients' tumors and blood (such as PET-CT parameters, tumor mutational signatures, RNA analyses, neoantigens, circulating tumor DNA, and T cell activation and exhaustion) should be evaluated in this and other ongoing neoadjuvant trials.

3. Failing to do 2, results in a paper where the main material is highlights of a few interesting cases amongst the 8 responders.

Reply: Thank you for your comment. As mentioned before, we have finished DNA analyses to improve the depth of translational analysis to some extent. And a subsequent trial with a more in-depth biomarker analysis is being planned (NCT05069857).

4. I am surprised to see that PD1 staining did not change after treatment with PD1-blockade? would there not be occupancy of the receptor preventing binding of the IHC antibody?

Reply: Thank you for your comment. Camrelizumab mainly utilizes its heavy chain to bind to PD-1 while the light chain sterically inhibits the binding of PD-L1 to PD-1 through the interaction of glycosylation of asparagine 58 (N58).¹ The binding sites of the immunofluorescence antibody (CST, PD-1 (D4W2J) XP® Rabbit mAb #86163, used in this trial) are residues surrounding Ala274 of human PD-1 protein. There is little chance that the occupancy of the receptor.

The purpose of staining we did was to explore if PD-1 protein expression could be used as a predictive biomarker rather than to test the block effect of the drug by immunofluorescence for PD-1 expression. Indeed, it would be interesting to detect the dynamics of PD-1 glycosylation or other markers through neoadjuvant

immunotherapy, which will be performed in our subsequent trial. Many thanks for your kind help!

Reference:

1. Liu K, et al. N-glycosylation of PD-1 promotes binding of camrelizumab. *EMBO Rep* 21, e51444 (2020).

5. while the angiogenic markers are changing, it does not appear to be affecting their correlation with response, is the antiangiogenic drug affecting other pathways or other cell types such as macrophages etc

Reply: Thank you for your comment. In this trial, we found that angiogenesis markers CD31 and α -SMA were both inhibited by neoadjuvant therapy; however, the baseline expression of CD31 and α -SMA showed no differences between the MPR and non-MPR groups. We speculated that this confusing finding might be owing to the small sample size of the present trial. On the other hand, the multiple steps of the cancer immunity cycle and multiple signaling pathways have been found to be affected when combining the antiangiogenic agent with anti-PD-1 therapy. Anti-angiogenic therapy can reduce angiogenic signaling and result in normalization of the tumor vasculature to diminish the immunosuppression exerted by immunosuppressive cells (e.g. Tregs, tumor-associated macrophages, and myeloid-derived suppressor cells), as well as can inhibit the expression of PD-1 and regulate apoptotic pathways in cytotoxic CD8⁺ T cells, thereby improving the efficacy of immunotherapy.^{1,2} Huang et al. also reported that the crosstalk between tumor vascular normalization and immune reprogramming exists and plays a vital role in responses.³ In addition, the LT β R signaling pathway and angiogenic pathway ANGPT2/Tie2 have been reported to be modulated after combinatory treatment with an anti-angiogenic agent and anti-PD-1 antibody.⁴ In our trial, whether apatinib would affect other pathways or other cell types when combined with camrelizumab deserves further investigation. However, the baseline tumor tissue obtained by biopsy is inadequate for further exploration, which is indeed

a limitation.

The above information has been supplemented in the Discussion part (page 14, paragraph 2). Many thanks for your kind help!

Reference:

1. Tian L, et al. Mutual regulation of tumour vessel normalization and immunostimulatory reprogramming. *Nature* 544, 250-254 (2017).
2. Song Y, Fu Y, Xie Q, Zhu B, Wang J, Zhang B. Anti-angiogenic Agents in Combination With Immune Checkpoint Inhibitors: A Promising Strategy for Cancer Treatment. *Front Immunol* 11, 1956 (2020).
3. Huang Y, Kim BYS, Chan CK, Hahn SM, Weissman IL, Jiang W. Improving immune-vascular crosstalk for cancer immunotherapy. *Nat Rev Immunol* 18, 195-203 (2018).
4. Lee WS, Yang H, Chon HJ, Kim C. Combination of anti-angiogenic therapy and immune checkpoint blockade normalizes vascular-immune crosstalk to potentiate cancer immunity. *Exp Mol Med* 52, 1475-1485 (2020).

Minor issues that need to be addressed:

1. many typographical errors that need to be corrected, including those in the main figures (eg Fig 1- surgery and radiotherapy have been misspelled

Reply: We are very sorry for our incorrect writing. We have corrected the misspellings in Fig. 1.

2. missing legends for many of the graphs (eg Figure 2, 4, 5).

Reply: Thank you for your comment. The legends of Figure 2, 4 and 5 have been supplemented in the Figure legends (page 30 - 31, revised version). The detailed legends are as follows:

Fig. 2 Efficacy of neoadjuvant therapy. (A) Residual viable tumor cells (RVT) ratio, combined positive score (CPS), and radiographic partial response (PR) in 20 patients. The percentage of RVT was evaluated on resected tumor slides after surgery. The

CPS was defined as the total number of programmed cell death-ligand 1 staining cells (including tumor cells, tumor-associated lymphocytes, and macrophages) divided by the total viable tumor cells plus 100. Radiographic response according to RECIST 1.1 was performed on basis of imaging examinations before and after neoadjuvant therapy (green triangles for the MPR group [n = 8], black triangles for the non-MPR group [n = 12]). (B) In the patient achieved pathological complete response, images of the oral tongue (left) and magnetic resonance imaging (right) before (upper) and after (lower) neoadjuvant therapy.

Fig. 4 Features of the hyperprogression disease. In the No.14 patient who showed hyperprogression disease: radiographic and HE staining images of before- (A) and after- (B) neoadjuvant therapy. (C) Multiplex immunofluorescence images of the tumor site before- and after- neoadjuvant therapy. Primary antibodies targeting CD163, CD68, PD-1, CD8, PD-L1, and Pan-CK were used. Nuclei acids were stained with DAPI. (D) Comparison of CD8+ and CD163+ intensity. After staining, slides were scanned and multilayer images were used for quantitative image analysis. The quantities of various cell populations were expressed as the number of stained cells per square millimeter in all nucleated cells.

Fig. 5 Anti-angiogenesis evaluation. (A) Representative immunofluorescence staining images of before- and after- neoadjuvant therapy tumor sections (Case No.7, green for CD31, red for α -SMA, blue for DAPI). (B) Fluorescence intensity of CD31 and α -SMA between before- and after- neoadjuvant therapy tumor tissues in all 20 patients. Whiskers represent min to max. The significance for differences between before- and after- neoadjuvant therapy was tested using Wilcoxon signed-rank test. (C) Comparison of baseline CD31 and α -SMA fluorescence intensity between MPR and non-MPR groups (n = 8 in MPR group, n = 12 in non-MPR group). The significance for differences between MPR and non-MPR group was tested using Mann Whitney test. Bars represented mean with SD.

3. Statistical tests used should also be indicated in the figure legends.

Reply: Thank you for your comment. We have added the information of statistical

tests in the Figure legends of Figure 3 and 5 (page 30 -31, revised version). The detailed information is as follows:

Fig. 3 The significance for differences between before- and after- neoadjuvant therapy was tested using Wilcoxon signed-rank test; for differences between MPR and non-MPR group, the significance was tested using Mann Whitney test. Bars represented mean with SD.

Fig. 5 (B) Fluorescence intensity of CD31 and α -SMA between before- and after- neoadjuvant therapy tumor tissues in all 20 patients. Whiskers represent min to max. The significance for differences between before- and after- neoadjuvant therapy was tested using Wilcoxon signed-rank test. (C) Comparison of baseline CD31 and α -SMA fluorescence intensity between MPR and non-MPR groups (n = 8 in MPR group, n = 12 in non-MPR group). The significance for differences between MPR and non-MPR group was tested using Mann Whitney test. Bars represented mean with SD.

Reviewer #2 (Remarks to the Author):

I would like to commend the authors on a well put together manuscript. Below are a few minor comments/suggestions:

Reply: Thank you for your appreciation of our study. We have revised the manuscript according to your comments below.

1. For the study endpoints, are all AEs reported or only treatment related AEs?

Reply: Thank you for your comment. In this trial, we reported all treatment-related AEs of any grade and two SAEs which were considered unrelated to study treatment. The detailed description of safety in the manuscript is as follows (page 5, paragraph 3).

2. In the study design, is there a reference for the MPR rate of 7%? This should potentially be mentioned in the background as well.

Reply: Thank you for your comment. Before the initiation of our study, neoadjuvant pembrolizumab and nivolumab monotherapy have been explored in HNSCC and oral cancer, and achieved MPR rates of 4.3% (pembrolizumab for HNSCC) and 8% (nivolumab for OSCC), respectively. Based on this, we took a MPR rate of 7% as a reference for sample size calculation.

According to your comment, we added this part in the Background part (page 4, paragraph 2) as: In the neoadjuvant setting, immune checkpoint blockade has shown promising results against many other tumor types. However, for OSCC or HNSCC, neoadjuvant anti-programmed cell death-1 (PD-1) monotherapy has showed a relatively low major pathological response (MPR) rate (4.3% for pembrolizumab in HNSCC and 8% for nivolumab in OSCC)^{1,2}.

We also added this part in the Methods part (page 19, paragraph 1) as: Statistical analyses - A sample size of 20 evaluable patients was required to achieve 90% power to detect an increase in the MPR rate from 7% (anti-PD-1 monotherapy, based on data from pembrolizumab and nivolumab monotherapy^{1,2}) to 30%, with a one-sided exact test with a significance level (alpha) of 0.0500.

Reference:

1. Wise-Draper T, et al. 809 Phase 2 trial of neoadjuvant and adjuvant PD-1 checkpoint blockade in local-regionally advanced, resectable HNSCC indicates pathological response is associated with high disease-free survival. *Journal for ImmunoTherapy of Cancer* 8, A484-A485 (2020).
2. Schoenfeld JD, et al. Neoadjuvant Nivolumab or Nivolumab Plus Ipilimumab in Untreated Oral Cavity Squamous Cell Carcinoma: A Phase 2 Open-Label Randomized Clinical Trial. *JAMA Oncol* 6, 1563-1570 (2020).
3. When comparing the fluorescence using the t-test, was normality assessed? Many of the markers appear as though they may be skewed.

Reply: Thank you for your comment. After the normal distribution test of the data, we corrected the statistical methods in the Methods part (Page 19, Paragraph 2) and

figure legends part (Page 30 - 31) as: Differences between MPR and non-MPR group were analyzed using Mann Whitney test, and differences between before- and after-neoadjuvant therapy within a group were analyzed using Wilcoxon signed-rank test.

4. In figure 3, the color code does not match the legend (images have green/gray not blue/red).

Reply: We are very sorry for the mistake and inconvenience it caused in your reading. We have corrected the typographical errors in legend of figure 3 (page 30, paragraph 2).

4a. In figure 3, were comparisons made between pre-post within a group? It might also be worthwhile to compare the changes in these markers between groups.

Reply: Thank you for your comment. We have compared all markers between pre- and post- within a group and compared the changes in these markers between MPR and non-MPR groups. The corresponding results have been added into Fig 3, Supplementary Fig. S1, Supplementary Fig. S3, and the result part of the revised manuscript (page 8, paragraph 3) as: Multiplex immunofluorescence (MIF) for tumor-infiltrating lymphocytes (TILs) staining showed significant increases in CD68+CD163+ ($P = 0.04$) and CD8+/FoxP3+ ratio ($P = 0.002$) while decreases in CD3+ ($P = 0.03$) and FoxP3+ ($P = 0.03$) from before- to after- neoadjuvant therapy (Supplementary Fig. S1 in Additional file 4). The changes of all markers over the neoadjuvant therapy were compared between the MPR and non-MPR groups, but no significant differences were found (Supplementary Fig. S2 in Additional file 4). The characteristics of TILs infiltration in surgically resected tumors between two groups were further compared, no significant difference was found in CD8+ T cell infiltration (Fig. 3C) but the MPR group showed more CD4+ T cell infiltration ($P = 0.02$, Fig. 3D, Supplementary Fig. S3 in Additional file 4).

4b. Please provide the actual p-values for significant results.

Reply: Thank you for your comment. We have added actual p-values for all

significant results into the Result part and the figures.

5. if not all patients have adequate follow-up (ie. 12 months), then is the clopper-pearson method appropriate for estimating the 12-month survival and local recurrence rates? These should be estimated off of Kaplan-Meier curves.

Reply: Thank you for your comment. We have updated the information of follow-up and re-evaluated the survival rate and local recurrence rates using Kaplan-Meier method and added them into the Result part as (page 7, paragraph 4):

As of March 2022, the median follow-up time was 18 months (range 15–22 months). Two patients who did not receive adjuvant radiotherapy had contralateral neck lymph node metastasis and local recurrence, respectively. The estimated 18-month locoregional recurrence rate was 11.2% (95% CI: 3.8%–18.6%). One patient died, and the estimated 18-month overall survival rate was 95% (95% CI: 90.1%–99.9%).

We have added the corresponding description of the statistic method into the Method part (page 19, paragraph 2) as: The estimated overall survival rate and local recurrence rate were evaluated with the Kaplan-Meier method.

6. In the efficacy results, a confidence interval for the MPR rate should be provided, along with the exact test p-value comparing the observed rate to the null rate of 7%.

Reply: Thank you for your comment. We have added the confidence interval for the MPR rate and the exact test p value into the Result part (page 6, paragraph 2): The pathological efficacy indicated that MPR was observed in eight patients (40%, 95% confidence interval [CI]: 19.1%-63.9%). The MPR rate in this trial was statistically significantly higher compared to the null rate of 7% ($P = 0.00003$).

We have also added the corresponding description of the statistic method into the Method part (page 19, paragraph 2) as: The confidence intervals (CI) were calculated with the Clopper-Pearson method. The P value comparing the observed MPR rate to the null rate was calculated with the Fisher's exact test.

Reviewers' Comments:

Reviewer #1:

Remarks to the Author:

Thank you for addressing my comments and queries. As I stated before the biggest issue was always the question of how to position a study such as this. I believe that as a pilot study, it has great merit and appreciate how difficult it is to complete. The demonstrated response rates of 40% is remarkable and I also note the attempts made to further characterise using specific immune markers and also deep sequencing of a gene panel, which would provide important data on TMB and specific mutations. It would have been great if you also managed to obtain RNA-data (transcriptomics by RNAseq or nano string), but I understand your limitations and hope that these will be expanded upon in the future RCT that is being conducted.

Reviewer #3:

Remarks to the Author:

This reviewer has only two remaining concerns:

1. The Clopper-Pearson method for constructing confidence intervals (CIs) is still mentioned. This method is appropriate for complete data, such as the MPR rate, and for time-to-event data for time points at which there was no prior censoring. However, the authors present CIs for time points at which there appears to have been prior censoring (for example, 18 month overall survival with a range of follow-up times given as 15-22). I suspect an alternative method such as Brookmeyer-Crowley (Biometrics 1982) was used, but the authors should clarify.
2. I noticed a statement on p. 6-7 that "The number of patients with high CPS between the MPR and non-MPR groups showed significant differences ($P = 0.0005$, Pearson $r = 0.71$ for cutoff >10 ; $P = 0.004$, Pearson $r = 0.61$ for cutoff ≥ 20)." This would seem to be a comparison of two proportions, so it is not clear that a Pearson correlation coefficient is applicable.

Reviewer #1 - oral squamous cell carcinoma (immuno)therapy - (Remarks to the Author):

Thank you for addressing my comments and queries. As I stated before the biggest issue was always the question of how to position a study such as this. I believe that as a pilot study, it has great merit and appreciate how difficult it is to complete. The demonstrated response rates of 40% is remarkable and I also note the attempts made to further characterise using specific immune markers and also deep sequencing of a gene panel, which would provide important data on TMB and specific mutations. It would have been great if you also managed to obtain RNA-data (transcriptomics by RNAseq or nano string), but I understand your limitations and hope that these will be expanded upon in the future RCT that is being conducted.

Reply: Thanks for your comments and recognition. RNA analyses will be expanded upon in the randomized trial that is being conducted (NCT05069857).

Reviewer #3 - Replacement for Reviewer #2 - Clinical trials, biostatistician - (Remarks to the Author):

This reviewer has only two remaining concerns:

1. The Clopper-Pearson method for constructing confidence intervals (CIs) is still mentioned. This method is appropriate for complete data, such as the MPR rate, and for time-to-event data for time points at which there was no prior censoring. However, the authors present CIs for time points at which there appears to have been prior censoring (for example, 18 month overall survival with a range of follow-up times given as 15-22). I suspect an alternative method such as Brookmeyer-Crowley (Biometrics 1982) was used, but the authors should clarify.

Reply: Thank you for your comment. We double-checked the calculation of survival and recurrence rates, and have re-evaluated the CIs of survival and recurrence rates using Brookmeyer-Crowley method.

We have added all revisions into the Result part (page 7, paragraph 4) as: The estimated 18-month locoregional recurrence rate was 10.5% (95% CI: 0%–24.3%). One patient died, and the estimated 18-month overall survival rate was 95% (95% CI: 85.4%–100.0%).

We have added the corresponding description of the statistic method into the Method part (page 18, paragraph 3) as: The confidence intervals (CI) were calculated with Brookmeyer-Crowley method for survival and recurrence rates, and Clopper-Pearson method for MPR rate.

2. I noticed a statement on p. 6-7 that “The number of patients with high CPS between the MPR and non-MPR groups showed significant differences ($P = 0.0005$, Pearson $r = 0.71$ for cutoff >10 ; $P = 0.004$, Pearson $r = 0.61$ for cutoff ≥ 20).” This would seem to be a comparison of two proportions, so it is not clear that a Pearson correlation coefficient is applicable.

Reply: Thank you for your comment. We have re-evaluated the differences between two groups with Fisher's exact test, and revised the results (as shown in page 7, paragraph 1): The number of patients with high CPS between the MPR and non-MPR groups showed differences ($P = 0.004$ for cutoff >10 ; $P = 0.014$ for cutoff ≥ 20).

We have added the corresponding description of the statistic method into the Method part (page 19, paragraph 1) as: Based on the different CPS cutoffs, the P value of patient number between MPR and non-MPR groups was calculated with the Fisher's exact test.

Reviewers' Comments:

Reviewer #3:

None